# Betting Scenario for the Management of University Professional Practices from the Conformation of Intersectoral Cooperation Networks

Freddy Marín-González [1,*], Judith Pérez-González [2], Alexa Senior-Naveda [1], Mercy Narváez-Castro [3], Sharmila Rani Moganadas [4] and Eduardo Garcés-Rosendo [5]

1   Department of Humanities, Universidad de la Costa, Barranquilla 080002, Colombia
2   Department of Management, Universidad Nacional Experimental Francisco de Miranda, Coro 004102, Venezuela
3   Department of Postgraduate, Universidad del Zulia, Maracaibo 004102, Venezuela
4   Faculty of Business, Multimedia University, Melaka 75450, Malaysia
5   General Studies Division, Universidad Autónoma del Perú, Lima 015103, Peru
*   Correspondence: fmarin1@cuc.edu.co; Tel.: +57-3014056943

**Abstract:** The professional practices represent a space for interorganizational and intersectoral alliances that contribute to the development of localities and regions. From this referent, the design and validation of intersectoral cooperation networks is considered pertinent for the effective management of professional practices. Therefore, this article is oriented towards the construction of the ideal scenario where a universitygovernmentcompany intersectoral cooperation network can operate on a horizon of 2030. Foresight is used as a research and planning method, in conjunction with the consultation of experts from different social sectors that lead to obtaining sixteen scenarios of probable occurrence. For this, techniques, such as the prospective workshop, Delphi method, structural analysis (MICMAC) and scenario analysis (SMIC), are used. The results show a bet scenario where the four finally selected events occur, with a probability of occurrence of 35.7%, which would allow establishing future strategies that allow the network to be operational. It is concluded that the formation of a cooperation network for the management of university professional practices represents in itself a strategy to strengthen the curriculum and guide the achievement of common objectives in the intersectoral context studied. The contribution of the article to the study of sustainability sciences stands out, since it addresses a theme that leads to the description, explanation and understanding of the sustainable development of localities and regions from an educational dimension. In this sense, the contribution is synthesized from three planes of reflection and analysis: firstly, the understanding of sustainability as a multidimensional construct, where education is a key dimension to consolidate sustainable development processes; secondly, the management of interorganizational and intersectoral networks as a cooperation strategy that promotes sustainable development; and thirdly, prospective as a planning method that leads to delineating betting scenarios for sustainability management from an educational perspective, more specifically from the university curriculum.

**Keywords:** future; curriculum; prospective; higher education; educational internships

## 1. Introduction

Strategic prospective is considered by [1] as a formalized approach that uses a combination of qualitative and quantitative tools, to obtain multiple scenarios of future alternatives, supported by comprehensive analysis and incorporating possible actions and their consequences. In this regard, ref. [2] refer that it consists of analyzing the phenomena from a global, systemic and dynamic perspective, in order to generate future scenarios, built not only based on past data but also taking into account the future evolution of the variables, as well as the behavior of the actors involved, so as to minimize uncertainty and lead the

present action towards an acceptable or desired future. In the particular case of strategic prospective, scenarios are proposed as a tool that contributes to the analysis of opportunities and threats in the environment and thus describe possible futures in correspondence with a certain problematic situation [3] (p. 9). Noting that for [4] it is not enough that the foreseen futures can occur, it is also necessary that the social actors involved are able to make at least one of them a reality.

In this regard, ref. [5] refer that strategic prospective should not be understood as a prediction or forecast of tomorrow's events, but as an analysis of the future, to try to understand it and influence its dynamics The construction of scenarios reduces uncertainty regarding the future behavior of a certain social phenomenon, which allows for carrying out better planning and investment processes [6], in addition to facilitating the actors to participate in the construction of the desired future [7].

From the ideas exposed, it is inferred that the future can be estimated and thus provide a greater degree of certainty with respect to its trajectories of action; therefore, organizations seek to define and implement strategies to achieve the desired futures.

From the multidimensional perspective of sustainability, the creation of cooperation networks constitutes a key aspect of the social dimension associated with sustainable development, and in this, prospective is considered as a practical tool for strategic planning and management that enables processes of management of professional practices in higher education, as a space for exchange and interrelation of activity sectors that promote sustainability from intersectoral links. This way, this article is aligned with the educational trends associated with sustainability from the configuration of a higher education of the future.

The main purpose of the article is to build an ideal scenario where an intersectoral cooperation network can be operationalized that makes possible the effective management of professional practices developed by university students in their last years of academic training, intervening as an empirical context for research in the Paraguaná Peninsula, Falcón state, Venezuela.

Regarding the prospective to estimate the impact associated with the operation of an intersectoral cooperation network (university–company–student), it could contribute to universities developing improvements in their management mechanisms, educational quality processes, obtaining competitive advantages in their mission functions, strengthening relationships with their environment, as well as assessing the relevance of the profile of graduates, among other descriptors. In relation to companies, improvement in their characteristic processes and products, training of their employees is expected; likewise, innovation and solutions to context problems would be encouraged. Concerning students, they would show improvements in their training processes from a complex and transdisciplinary vision, based on theory–practice integration; the purpose is to build significant learning experiences that correspond to the dynamics of the world of work, as a basis for developing professional and occupational skills.

In this regard, ref. [8] state that the importance of cooperation lies in the fact that it contributes to reinforcing the competitive position of companies and, consequently, of their territories. In this same order, in [9] they refer to the improvement of the standard of living of localities and rural or urban territories from the cooperation between the social sectors.

From the cooperation networks, a greater increase in the accessibility of resources, a greater amount of knowledge and shared information, the creation of new product lines and a greater competitive consolidation are promoted [10]. In this regard, ref. [11] emphasizes the transfer of knowledge as a challenge for educational organizations.

The consolidation of networks is inferred as a way to strengthen transaction and exchange processes. Taken into consideration [12], where they consider that the driving factors of network cooperation multiply as a network evolves in time and space, and the factors that influence it become increasingly diverse. In this sense, authors, such as [13,14], highlight the importance of educational prospective and network design, as key strategic processes for growth and development in organizational, interorganizational, and intersectoral areas.

Consequently, in this article, there is a search for future places for the reader from the dynamics of the intersectoral context, which is the object of the investigation, then the methodological systematization is presented to later organize the results, their analysis, and interpretation in light of the theoretical references.

## 2. Background of the Study

Numerous investigations have been carried out with the purpose of studying the activities of companies in cooperation networks in coherence with the ideas of [15,16]; the competences of social innovation agents, to operate jointly, require that the organizations and among them universities generate intersectoral spaces that are configured in cooperation networks around common objectives capable of operationally connecting the different sectors of activity in society and specifically in the context of management of professional practices of the students [17], which occur mainly in the productive sector [18]. This relationship that arises in the academic context generates strong and lasting links as well as alliances among the various sectors that are intertwined in the training processes of professional practice in universities, enabling the formation of intersectoral cooperation networks [16].

In coherence with the perspective of [19] who developed research on the organizational DNA for the quality of service in public universities, they affirm that the development of human talent must be oriented to the performance of alternatives and innovations in the different universities from the production and application of knowledge. From the perspective of [9], the creation of intersectoral knowledge networks favors the development of the productive, government and university sectors, reduces uncertainty and promotes the social capital of the locality, which contributes to decision making in the various sectors of activity, while strengthening relationships between them [20].

This finding is consistent with the results of [21], interaction between cultural, social, economic and political factors leads to the emergence of complex problems [22]. A vast amount of evidence calls for a new solutions that cannot be found by a single actor from one sector; in this sense, it states that the interactions and innovations in the university, government and productive sector spheres behave like a triple helix [23,24] that opposes creative knowledge gaps to solve problems in the government or productive sector, a path different from the linear technocratic one, but rather a disruptive path that involves alliances and cooperative actions in the network as a space where innovations are generated to solve real problems in different sectors of society [25,26], as well as the promotion of the establishment and/or consolidation of strategic ties between governments, the productive sector, civil society organizations and institutions of university education, science and technology [27,28].

From these perspectives, management in higher education is assumed as an intentional process organized and oriented towards the optimization of operations, procedures and internal projects of higher education institutions, with the aim of approaching the purposes for which they were created. For the institutions, which implies perfecting the pedagogical, managerial, community and administrative procedures that are mobilized in it, to achieve this, it is necessary to have the competition and joint work of all interested parties [29,30].

The actors in the educational field, to obtain successful and innovative results, require strengthening of the capacities to cooperate and establish lasting alliances and agreements that make it possible to project, design, analyze and evaluate the processes of labor and disciplinary training of students in spaces of professional practice, so that the construction and transfer of knowledge that occurs in academic and work spaces can become innovations that generate social welfare to the extent that they respond to the demands of the environment [31].

In this framework of ideas, the management processes of education systems require attention to factors, such as planning, equity, quality, resource management, community participation through a good governance model that guarantees the governance and governability of educational institutions, generating trust and credibility from accountability to

society. All of this is to generate optimal results and the provision of better services; among them, the management of professional practices.

## 3. Research Context

It is worth noting that a university–government–business cooperation network (CUGE network) is proposed, with a local radius of action in the Paraguaná Peninsula, which is located north of Falcón state, in Venezuela. The Paraguaná Peninsula is made up of three municipalities: Carirubana, Los Taques and Falcón. It is a xerophilous zone of arid soil, if in its entirety, with a territorial extension of 3405 km$^2$ and an approximate population for the year 2020 according to the National Statistics Institute (INE) [32] of 292,427 inhabitants for the Carirubana municipality, 58,034 inhabitants for the Falcón municipality, and 48,229 inhabitants for the Los Taques municipality; the estimated population in the Peninsula being 398,690 inhabitants, representing 36.46% of the total population of the state.

The main economic activity of the state is related to the oil industry. Two refineries with large production capacity for export and consumption throughout the country are located in the Paraguaná Peninsula, which comprise the so called Paraguaná Refining Complex (PRC).

Likewise, in the previous investigative approach, the internal and external analysis of the participating sectors was carried out with the strengths–weaknesses–opportunities–threats (SWOT) matrix, in order to define strategies aimed at reducing weaknesses, enhancing strengths, neutralizing threats, and using the opportunities present in the context of the network; the identified strategies contribute to the optimal functioning of the proposed network. In addition, the analysis of the attributes by sectors of the participating organizations was carried out in terms of their activity, capacity, trajectory and positioning in order to later be able to characterize the competency profile of each actor, related to their role and participation within the network.

## 4. Materials and Methods

Foresight through the scenarios method comprises two phases: the construction of the analytical base and the elaboration of scenarios that lead to the establishment of forecasts considering driving factors, trends, strategies of the actors and the terms of change studied [33]. There is no universal pattern or single methodology for configuring desirable, probable, and possible scenarios [34]. Specifically, in this article, the scenario method is partially applied, integrating principles of relevance, coherence, probability, importance and transparency [35], for which a 10-year time horizon of analysis is considered (estimate to 2030), see Figure 1.

However, in order to search for the ideal scenario, in which an intersectoral university–government–company cooperation network can operate, it is based on the theoretical foundation that explains the dynamics of networks in the intersectoral spaces identified in relation to the management of professional practices. With respect to the empirical component, the research was located in the Paraguaná Peninsula, Falcón state, Venezuela, a geographical space characterized by oil refining and important industrial activity associated with the tourist and commercial investment-free zone. In this sense, we work with the survey technique through a structured questionnaire that allows us to anticipate the occurrence of events associated with hypotheses related to the design of a cooperation network.

The key stakeholders who participated in the study are identified below:

A representative of the different chambers (business groups) in the area; namely, the Oil Chamber, the Chamber of Industry, the Chamber of Commerce, and the Chamber of Construction.

The director of professional practices of the public universities of the Paraguaná Peninsula, Falcón, Venezuela, represented in the study by the University of Zulia (LUZ), National Experimental University "Francisco de Miranda" (UNEFM), University Liberator Experimental Polytechnic (UPEL), Bolivarian University of Venezuela, Falcón Extension

(UVB), Polytechnic University of the Armed Forces, Punto Fijo Branch (UNEFA), and National Open University (UNA).

**Figure 1.** Scenario method [35].

The representatives of the regional government are made up of a representative of the Mayor's Offices of the Carirubana, Los Taques and Falcón municipalities, in addition to a representative of the Falcón state government. That is, the empirical moment and the theoretical foundations of the research were contrasted, which allowed revealing the operational variables of the proposed network, in the understanding that prospective evaluation requires descriptive mechanisms of the reality of the context, that is, go through the search for information, define needs, data, alternative courses of action as a basis for the projection of scenarios. Likewise, at this stage it is possible to identify the key actors.

From a multi-method and complementarity vision, auxiliary techniques are applied, such as prospective workshop, consensus, Delphi method, MICMAC® prospective software (Cross Impact Matrix-Multiplication Applied to a Classification), SMIC survey, Smic-Prob Software Expert probabilized cross impacts) which, together with the consultation of seven experts from different areas (business, university, political and social), and in coordination with the prospective team, allowed us to determine the possible scenarios in which the proposed network can operate.

## 5. Results

### 5.1. Structural Analysis

Structural analysis provides the most exhaustive possible representation of the system and allows its complexity to be reduced to the essential variables, which is equivalent to highlighting the key variables of the system, whether they are hidden [36]. It leads to listing and identifying the variables associated with the design of the network. The step prior to the establishment of relationships between them consisted in defining each variable, with the objective that the actors participating in the analysis appropriated its conception and meanings. The detailed explanation of the variables facilitates the location of their relationships, which constitutes the foundation of necessary topics for any prospective reflection [35].

At this stage, after its identification and definition of the prospective team, the variables were regrouped and dimensioned in three large organizational contexts, as shown in Table 1.

**Table 1.** Sizing of the variables.

| Dimension | Variable |
|---|---|
| Professional internship programs in the university sector | $X_1$: Legal aspects related to professional practices |
| | $X_2$: Administrative processes that support professional practices |
| | $X_3$: Labor relations that are carried out in the exercise of professional practices |
| | $X_4$: Formative contents covered in professional practices |
| | $X_5$: Didactic orientation for the execution of practices |
| | $X_6$: Personal relationships between the actors involved |
| Intersectoral cooperative relations. | $Y_1$: Trust and commitment between actors |
| | $Y_2$: Information flow between actors |
| | $Y_3$: Shared goals |
| | $Y_4$: Geographical proximity |
| | $Y_5$: Regulatory flexibility of the actors involved |
| | $Y_6$: Participation of the actors in the construction of the graduation profile of university students |
| | $Y_7$: Relevance of undergraduate training, according to the needs of the government and business sector |
| | $Y_8$: Infrastructure and equipment for networking |
| | $Y_9$: Previous relationships with the actors involved |
| | $Y_{10}$: Governance system |
| | $Y_{11}$: Reticular culture |
| | $Y_{12}$: Management support |
| Intersectoral scope university–government sector– business sector | $Z_1$: Execution of the university–government–company relationship based on the management of professional practices |
| | $Z_2$: Modality of university–government–company relationship (linkage, association, cooperation) |
| | $Z_3$: Intersectoral alliances |

It is highlighted that the study variables emerged from the transit of a descriptive diagnosis in relation to the input knowledge of existing theories, such as social network theories, cooperation theories, and university relationship models with the external sector (company, government), as well as the university curriculum regarding the management of professional practices. The treatment that some authors make to certain categories considered mainly in the investigation was revealed, in this sense, scientific articles, books, university

curriculum, and doctoral theses in relation to the investigated topic were analyzed, which allowed the generation of a questionnaire of closed questions with a Likert-type scale, which was applied to the key actors of the study, prior to a validation process through expert judgment and reliability determined through Cronbach's Alpha coefficient.

### 5.2. Analysis of the Variables According to Their Motor Skills and Dependency

Once the variables were listed and defined, the relationship between each one of them was described and in correspondence with the system of which they are a part, where relationship forces converge and diverge, vary depending on the intensity or type of variable, so it is necessary to know the incidence that the rest has on one of them, that is, to demonstrate the existence of the variables only by their relational weave with other variables [35].

This analysis was carried out taking the opinion of experts as a referential framework (seven experts in total) through the motor dependency matrix; it was also considered: (a) that a member of the prospective team play the role of moderator of the group and facilitator of the information to execute the session, and (b) evaluate the existence of influences between variables; in this case, the value of one (1) is assigned for the existence of a relationship and zero (0) when no such relationship is perceived. The aforementioned is shown in Table 2.

**Table 2.** Structural analysis matrix.

| Influence of/on | Direct Influence | | | | | | | | | | | | | | | | | | | | | Motor Skills |
|---|---|---|---|---|---|---|---|---|---|---|---|---|---|---|---|---|---|---|---|---|---|---|
| | $X_1$ | $X_2$ | $X_3$ | $X_4$ | $X_5$ | $X_6$ | $Y_1$ | $Y_2$ | $Y_3$ | $Y_4$ | $Y_5$ | $Y_6$ | $Y_7$ | $Y_8$ | $Y_9$ | $Y_{10}$ | $Y_{11}$ | $Y_{12}$ | $Z_1$ | $Z_2$ | $Z_3$ | |
| $X_1$ | | 1 | 1 | 0 | 0 | 1 | 0 | 1 | 1 | 0 | 1 | 1 | 1 | 1 | 0 | 1 | 1 | 1 | 1 | 1 | 1 | 15 |
| $X_2$ | 1 | | 1 | 0 | 0 | 1 | 0 | 1 | 1 | 0 | 1 | 1 | 1 | 0 | 1 | 0 | 1 | 0 | 1 | 1 | 1 | 13 |
| $X_3$ | 0 | 1 | | 0 | 0 | 1 | 1 | 1 | 1 | 0 | 1 | 1 | 1 | 1 | 1 | 1 | 1 | 1 | 1 | 1 | 1 | 16 |
| $X_4$ | 0 | 1 | 0 | | 1 | 1 | 1 | 1 | 1 | 0 | 0 | 1 | 1 | 1 | 1 | 0 | 0 | 1 | 1 | 1 | 1 | 14 |
| $X_5$ | 0 | 1 | 1 | 1 | | 1 | 1 | 1 | 1 | 0 | 0 | 1 | 1 | 1 | 1 | 1 | 1 | 1 | 1 | 1 | 1 | 17 |
| $X_6$ | 0 | 0 | 1 | 1 | 1 | | 1 | 1 | 1 | 0 | 1 | 1 | 1 | 1 | 1 | 1 | 1 | 1 | 1 | 1 | 1 | 17 |
| $Y_1$ | 1 | 1 | 1 | 0 | 1 | 1 | | 1 | 1 | 0 | 0 | 0 | 0 | 0 | 1 | 1 | 1 | 1 | 1 | 1 | 1 | 15 |
| $Y_2$ | 1 | 1 | 1 | 1 | 1 | 1 | 1 | | 1 | 0 | 1 | 1 | 1 | 1 | 1 | 1 | 1 | 1 | 1 | 1 | 1 | 19 |
| $Y_3$ | 1 | 1 | 1 | 1 | 1 | 1 | 1 | 1 | | 0 | 1 | 1 | 1 | 1 | 1 | 1 | 1 | 1 | 1 | 1 | 1 | 19 |
| $Y_4$ | 0 | 0 | 1 | 0 | 0 | 1 | 0 | 0 | 1 | | 0 | 0 | 0 | 1 | 1 | 0 | 1 | 1 | 1 | 1 | 1 | 10 |
| $Y_5$ | 1 | 1 | 1 | 0 | 0 | 1 | 1 | 1 | 1 | 0 | | 0 | 0 | 0 | 0 | 1 | 1 | 1 | 1 | 1 | 1 | 13 |
| $Y_6$ | 1 | 1 | 1 | 1 | 1 | 1 | 1 | 1 | 1 | 0 | 1 | | 1 | 1 | 1 | 1 | 1 | 1 | 1 | 1 | 1 | 19 |
| $Y_7$ | 1 | 0 | 1 | 1 | 1 | 1 | 1 | 1 | 1 | 0 | 0 | 1 | | 1 | 1 | 1 | 1 | 1 | 1 | 1 | 1 | 17 |
| $Y_8$ | 0 | 1 | 1 | 1 | 1 | 1 | 1 | 1 | 1 | 1 | 0 | 1 | 1 | | 1 | 1 | 1 | 1 | 1 | 1 | 1 | 18 |
| $Y_9$ | 0 | 1 | 1 | 1 | 1 | 1 | 1 | 1 | 1 | 1 | 1 | 1 | 1 | 1 | | 0 | 1 | 1 | 1 | 1 | 1 | 18 |
| $Y_{10}$ | 1 | 1 | 1 | 1 | 0 | 1 | 1 | 1 | 1 | 0 | 1 | 1 | 1 | 1 | 1 | | 1 | 1 | 1 | 1 | 1 | 18 |
| $Y_{11}$ | 1 | 1 | 1 | 0 | 1 | 1 | 1 | 1 | 1 | 0 | 1 | 1 | 1 | 1 | 1 | 1 | | 1 | 1 | 1 | 1 | 18 |
| $Y_{12}$ | 1 | 1 | 1 | 1 | 1 | 1 | 1 | 1 | 1 | 0 | 0 | 1 | 1 | 1 | 1 | 1 | 1 | | 1 | 1 | 1 | 18 |
| $Z_1$ | 1 | 1 | 1 | 1 | 1 | 1 | 1 | 1 | 1 | 0 | 1 | 1 | 1 | 1 | 1 | 1 | 1 | 1 | | 1 | 1 | 19 |
| $Z_2$ | 0 | 1 | 1 | 1 | 1 | 1 | 1 | 1 | 1 | 1 | 1 | 1 | 1 | 1 | 1 | 1 | 1 | 1 | 1 | | 1 | 19 |
| $Z_3$ | 0 | 1 | 1 | 1 | 0 | 1 | 1 | 1 | 1 | 0 | 1 | 1 | 1 | 1 | 1 | 1 | 1 | 1 | 1 | 1 | | 17 |
| Dependencia | 11 | 17 | 19 | 13 | 13 | 20 | 17 | 19 | 20 | 3 | 13 | 17 | 17 | 18 | 16 | 19 | 19 | 20 | 20 | 20 | | |

However, in Table 2, the sum of the numbers by rows is observed, which represents the times that each of the variables affects the rest. In this sense, $X_1$: legal aspects related to professional practices, affects or influences 15 of the remaining variables, in the same way that $X_2$: administrative processes that support professional practices influences 13 variables.

It follows that 15 and 13 represent the motor value of the variables $X_1$ and $X_2$. The number of variables on which a certain variable influences, or the percentage of influence of each variable, is called the motor index, that is, the influence or force that each variable has on the others; consequently, the greater the motor, the more influential, and the lower motor skills are less influential [37].

Likewise, it can be observed in Table 2 that the sum by columns represents the times that each variable is influenced by the rest; in this regard, $X_1$: legal aspects related to professional practices, is influenced by 11 variables, in the same way $X_2$: administrative processes that support professional practices, receives the influence of 17 variables. In this sense, the values 11 and 17 are called the dependency index, as they reflect the percentage of subordination of each variable.

In general, it is evident in the column that reflects the sum of motor skills, values in a range of 13 to 19, and the row with the sum of dependency presents values in a range of 11 to 20. In Table 3, it shows the relationship of the motor index of each variable with its corresponding dependency index.

**Table 3.** Relationship of motor indices—dependency by variable.

| Variable Symbology | Dependency | % | Motricity | % |
|---|---|---|---|---|
| $X_1$: Legal aspects related to professional practices | 11 | 3.151 | 15 | 4.297 |
| $X_2$: Administrative processes that support professional practices | 17 | 4.871 | 13 | 3.724 |
| $X_3$: Labor relations that are carried out in the exercise of professional practices | 19 | 5.444 | 16 | 4.584 |
| $X_4$: Formative contents covered in professional practices | 13 | 3.724 | 17 | 4.871 |
| $X_5$: Didactic orientation for the execution of practices | 13 | 3.724 | 17 | 4.871 |
| $X_6$: Personal relationships between the actors involved | 20 | 5.730 | 17 | 4.871 |
| $Y_1$: Trust and commitment between actors | 17 | 4.871 | 15 | 4.297 |
| $Y_2$: Information flow between actors | 19 | 5.444 | 19 | 5.444 |
| $Y_3$: Shared objectives | 20 | 5.730 | 19 | 5.444 |
| $Y_4$: Geographical proximity | 3 | 0.859 | 10 | 2.865 |
| $Y_5$: Regulatory flexibility of the actors involved | 13 | 3.724 | 13 | 3.724 |
| $Y_6$: Participation of the actors in the construction of the graduation profile of university students. | 17 | 4.871 | 19 | 5.444 |
| $Y_7$: Relevance of undergraduate training, according to the needs of the government and business sector. | 17 | 4.871 | 17 | 4.871 |
| $Y_8$: Infrastructure and equipment for networking | 18 | 5.157 | 18 | 5.157 |
| $Y_9$: Previous relationships with the actors involved | 18 | 5.157 | 18 | 5.157 |
| $Y_{10}$: Governance system | 16 | 4.584 | 18 | 5.157 |
| $Y_{11}$: Reticular culture | 19 | 5.444 | 18 | 5.157 |
| $Y_{12}$: Management support | 19 | 5.444 | 18 | 5.157 |
| $Z_1$: Execution of the university–government–company relationship depending on the management of professional practices. | 20 | 5.730 | 19 | 5.444 |
| $Z_2$: Modality of relationship established university–government–company | 20 | 5.730 | 19 | 5.444 |
| $Z_3$: Intersectoral alliances | 20 | 5.730 | 17 | 4.871 |
| TOTAL | 349 | 99.99% | 349 | 100% |

The generated data, as products of the interrelation of the variables, in terms of dependency and influence for the system under study, are represented on a Cartesian plane, where the X axis corresponds to dependency and the Y axis corresponds to influence. The purpose of the direct dependency motor plane is to visualize the dispersion of the variables and reflect on the participation in the system according to its position in the plane and which in turn allows the identification of several categories, namely input, repeater, excluded, resultant and platoon variables [38]; such variables are organized in turn into key, objective, first order regulatory, brakes or motors, autonomous, output and result variables. In Figure 2, the motor skills and direct dependency plan are presented. This plan gives a vision of the system based on the motor skills or dependence of the variables.

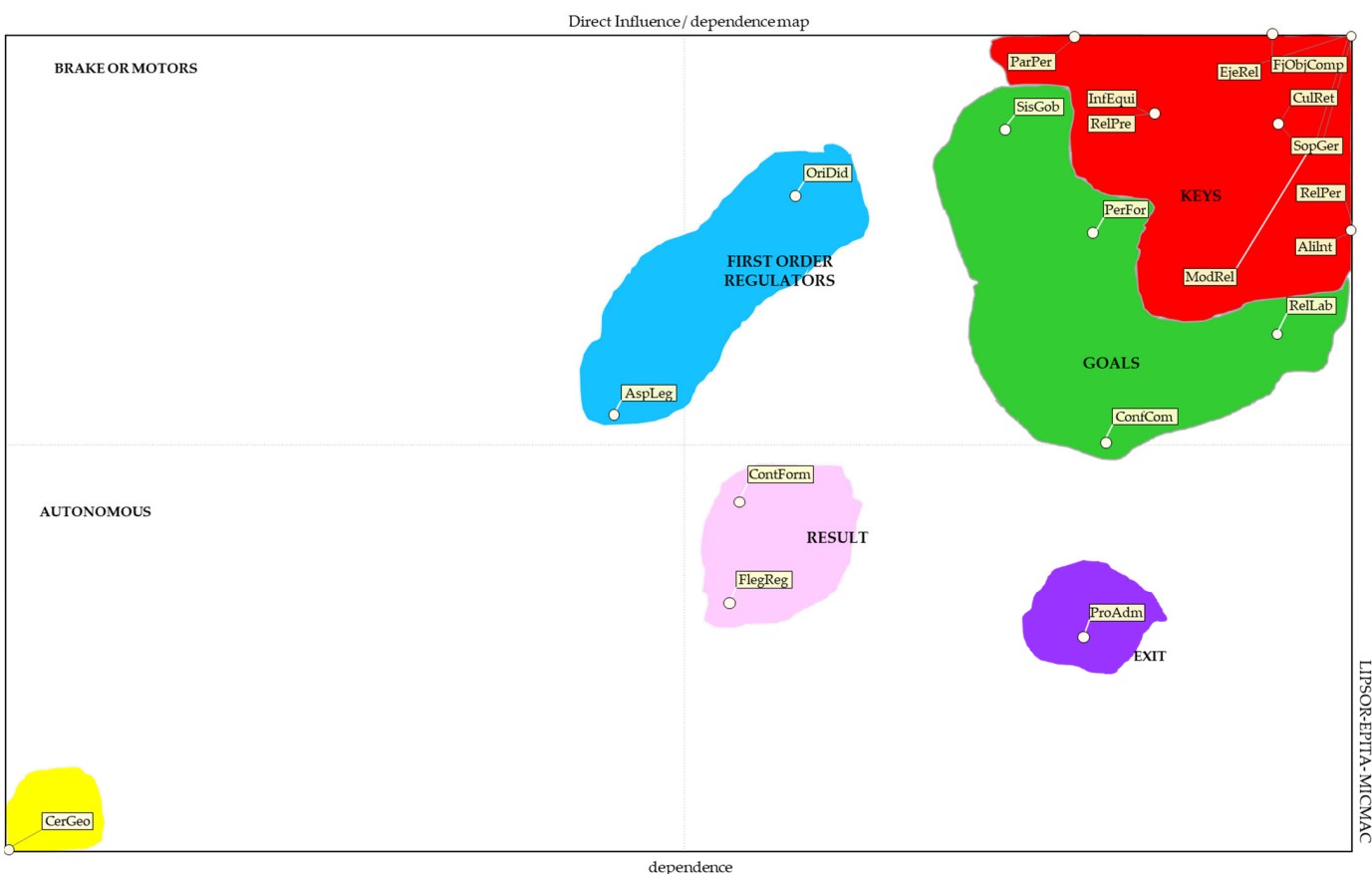

**Figure 2.** Motor skills–direct dependence and typology of variables. MICMAC® Program.

In attention to what was stated by [37], a location plan of the variables can be distinguished as indicated below: power zone, autonomous zone, conflict zone and exit zone. In this regard, the power zone is characterized by high motricity and the lowest dependency; consequently, the variables found in this zone are the most important in the system since they influence the vast majority and depend little on them. Any modification that occurs in it affects the entire system; the autonomous zone hosts the excluded variables, those that neither influence others nor are they influenced by the other variables of the system; the conflict zone, receives its name due to the great instability shown by the variables that converge there, due to its high mobility and high dependency; the exit zone, characterized by having low mobility but high dependency, groups the so-called result variables, as a consequence of the impact of the influence of the variables of the zone of power and conflict. See Figure 3.

| ZONE OF POWER $X_1$: Legal aspects related to Professional Practices. | CONFLICT ZONE $X_3$: Labor relations that are carried out in the exercise of professional practices, $X_5$: Didactic orientation for the execution of practices, $X_6$: Personal relations between the actors involved, $Y_1$: Trust and commitment between actors, $Y_2$: Flow of information between actors, $Y_3$: Shared objectives, $Y_6$: Participation of the actors in the construction of the graduation profile of university students, $Y_7$: Relevance of undergraduate training, according to the needs of the government and business sector, $Y_8$: Infrastructure and equipment to work in a network, $Y_9$: Previous relationships with the actors involved, $Y_{10}$: Governance System, $Y_{11}$: Reticular Culture, $Y_{12}$: Management support, $Z_1$: Execution of the university-government-company relationship - depending on the management of the professional practices, $Z_2$: Modality of relationship established between the university, the government and the company, $Z_3$: Intersectoral Alliances |
|---|---|
| AUTONOMOUS ZONE $Y_4$: Geographical proximity | STARTING AREA $X_2$: Administrative processes that support professional practices, $X_4$: Training content addressed in professional practices, $Y_5$: Regulatory flexibility of the actors involved |

**Figure 3.** Zone location plan.

*5.3. Identification of Key Variables*

Based on the information presented in the motor skills–dependency plans and in the area location plan, it was possible to identify six (6) important groups of variables in the system, namely:

1.  Input variables, determinants, brake or motors (above, on the left): they are highly motor, little dependent and determine the operation and evolution of the system. Within this group are legal aspects related to professional practices. Any modification in them will have repercussions throughout the system and determine its operation; depending on its evolution, they can become inhibitors or promoters of it. In the area of planning, for the decision makers, it is hoped that they stimulate the appropriate behaviors to follow for the improvement of the system under study.

2.  Repetitive variables (top right) subdivided into: key and objective variables. The keys are located in the upper right part, they are highly motor and dependent, they affect the normal functioning of the system, which makes them variables of great importance and members of the strategic axis; in this group we can mention: shared objectives, reticular culture, information flow, managerial support, execution of the university–government–company relationship, infrastructure and equipment for networking, among others. These variables have the capacity to disturb the normal functioning of the system. Any modification to them will have an effect on themselves and on the output variables.

The objective variables in this classification are found to be: labor relations that are carried out in the exercise of professional practices, relevance of undergraduate training according to the needs of the environment, governance system, trust and commitment of the actors, and training content covered in practices.

Note that these variables correspond to the intersectoral cooperation relationship dimension, one of the fundamental aspects of the network proposal to carry out university internships, they have a direct influence on their execution, therefore, they exercise action on the key variables. Figure 3 shows that these variables are located in the central part, are moderately motor and highly dependent, the level of dependency is the reason that gives them their name as objective variables. Its importance lies in the fact that it is possible

to act directly on them to promote their evolution, an important aspect to consider when making decisions about the proposed network.

3. Squad or first-order regulatory variables: located in the central part of the plane, they participate in the functioning of the system under normal conditions. They become "passing mechanisms" to achieve compliance with the key variables and make them evolve in a convenient way to achieve the objectives. They require good operation, so as not to cause a break in the studied system. Among them are legal aspects related to professional practices and didactic guidance for the execution of practices. To the extent that the legal aspects are flexible and there are didactic guidelines for the development of professional practices, the process will be effective in the actors involved. Given the variables identified here, it is necessary to monitor them periodically to avoid system breakdown.

4. Excluded or autonomous variables (below, on the left): they are low motor and dependent, so they are located in the area close to the origin. They correspond to past trends or inertia of the system, or they are disconnected from it. They do not constitute a determining part for the operation of the system. In this group, for the system under study, only geographic proximity is found. However, the geographical situation of the actors involved in the network must be taken into consideration, since at the historical moment when network design is proposed, it has great value, because in situations where a frontal relationship is warranted it could be hindered; likewise, the experts opted for the use and management of information and communication technologies to be in contact, without considering geographical proximity as something that interferes in the relationship.

5. Output variables (below, on the right): they account for the results of the operation of the system, they are little motor and highly dependent, they are classified as sensitive variables, often being translated as the objectives to be achieved, for which requires follow-up to control their performance and effectiveness on the studied system.

These variables cannot be addressed directly, but through the other variables of the system. In the study, these variables are training content addressed in professional practices, regulatory flexibility of the actors involved, and the administrative processes that support professional practices.

Most of these variables are associated with the professional practice programs dimension of the university sector; the administrative processes that support the practices support other processes in the network; the formative contents are of vital importance so that the students obtain answers to their needs and cover their expectations; the regulatory flexibility of the actors is also important, thus avoiding rigidity, bureaucracy and loss of time in the execution of internships. On the other hand, if the system does not take these aspects into account, the flight of students seeking to carry out their professional practices through the proposed network would begin.

*5.4. The Strategy Axis*

Once the analysis of the variable based on its location in the plan is complete, the next step is the definition of the strategic axis. On this axis it is located, in descending order, variables with a level of motor skills such that they become decisive variables for the functioning of the system, with a dependency that makes them susceptible to being addressed.

Figure 4 shows how the so-called key variables or integral variables of the strategic axis are grouped in the rear right part of the coordinate plane.

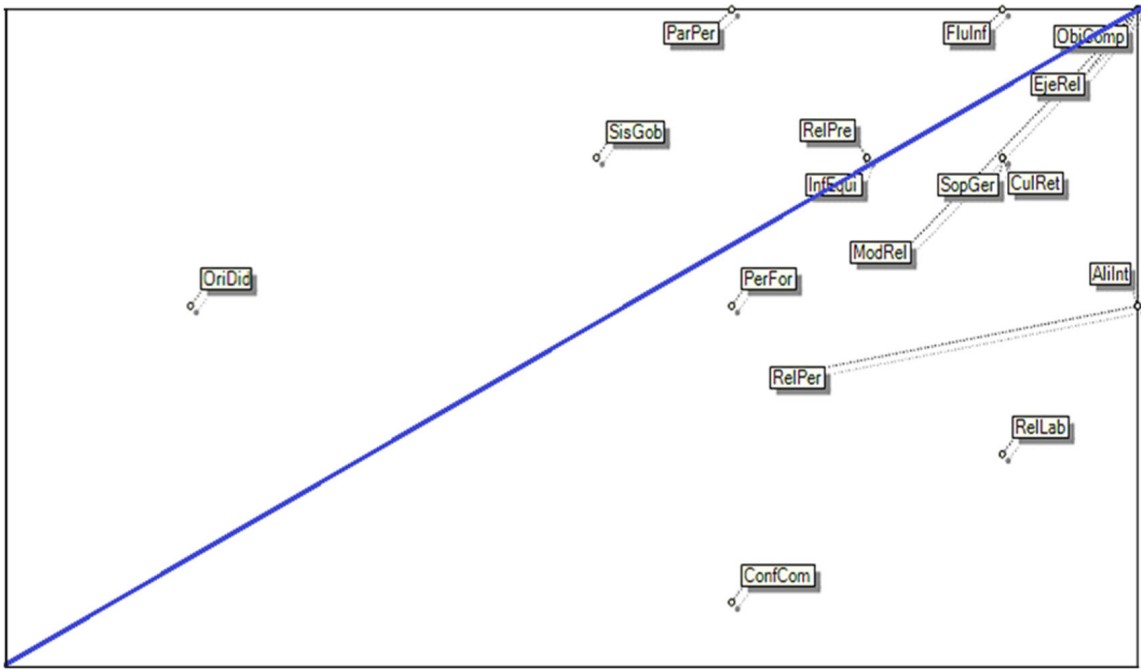

**Figure 4.** Location of the variables in the strategic axis. MICMAC® Program.

This is how the axis of the strategy appears represented as explained in the Zulia 2040 Strategic Prospective Plan of the Institute of Management and Strategy of Zulia, Venezuela (IGEZ) [39], by the projection of the cloud of variables plotted on an imaginary bisector, which, starting from the origin, is oriented towards the opposite vertex, offers an image of the layout of the main strategic challenges of the system, called so because their driving value in the future evolution of the system, is joined by a value of significant dependency for its approach.

Next, Figure 5 shows the hierarchy of the key variables in the strategic axis to be taken into consideration for the implementation of the proposed network.

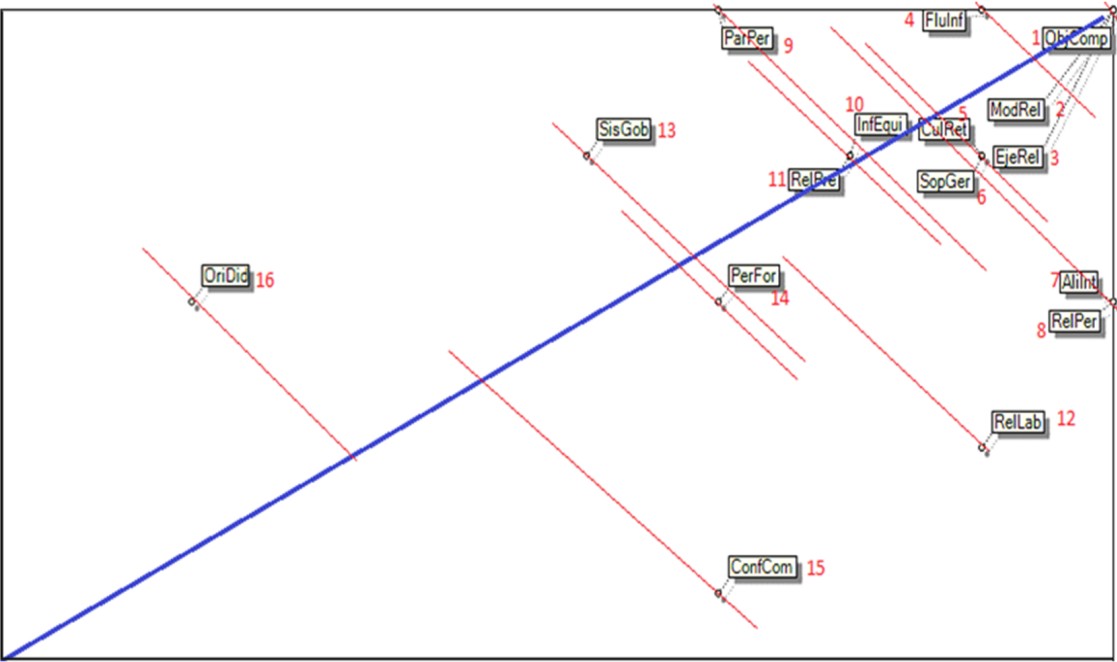

**Figure 5.** Strategic axis for the implementation of the CUGE network.

*5.5. Analysis of the Studied System Based on the Interpretation of the Arrangement of the Variables in the Axis of the Strategy*

A variable located between the keys or between the objective variables (among the most driving ones), could undoubtedly descend in the hierarchy in the axis of the strategy [40]. This is equivalent to saying that there are other variables that may be less of a driving force in the system, but that their actions on them will be more important, due to their ability to generate a greater number of actions in an eventual strategic plan.

As observed in the strategic axis, the system studied is characterized by being dynamic, unstable, and complex, due to the fact that the key variables, due to their proximitin the location of one with respect to the others, make their intervention difficult. They are in turn of high motor and dependent, internal to the studied system; the first ten variables of the strategic axis correspond entirely to the intersectoral cooperation relationship dimension, which implies solid relationships for the system to work. The weight of legal aspects of professional practices are promoters or inhibitors of this type of system. The study of variables through structural analysis complements decision making, increasing its effectiveness, from the generation of strategic information given by the variables of the studied system [41].

*5.6. Construction of Scenarios That Enable the Operation of the CUGE Network SMIC*

For the construction of scenarios, the key variables resulting from the structural analysis (direct and indirect relationships) are considered, in accordance with the position of the strategic axis identified below.

$Y_3$: Shared goals (conflict)

$Z_1$: Execution of the university-government–company relationship—depending on the management of professional practices (conflict)

$Z_2$: Modality of relationship established university–government–company (conflict)

$Y_2$: Information flow between actors (conflict)

$Y_{11}$: Reticular culture (conflict)

$Y_{12}$: Management support (conflict)

$Z_3$: Intersectoral alliances (conflict)

$X_6$: Personal relationships between the actors involved (conflict)

$Y_6$: Participation of the actors in the construction of the graduation profile of university students (conflict)

$Y_8$: Infrastructure and equipment for networking (conflict)

$Y_9$: Previous relationships with the actors involved (conflict)

$X_1$: Legal aspects related to professional practices (can)

Likewise, the analysis, argumentation and discussion of the researchers with the team of experts was considered, for which not only the interrelation of the variables was taken, but also the context where they interact (government sector–business sector–university sector); four (4) probable occurrence hypotheses were developed to guide the formulation of scenarios and thus achieve a more reliable and probable approximation of the scenario for the start-up of the CUGE network. In this sense, the identified hypotheses are described:

**H$_1$.** *The previous relationships between the university, the government, and the company, as well as the flow of information between them, will allow the formation of intersectoral alliances for the management of professional practices.*

**H$_2$.** *The modalities of relationship (linkage, association or cooperation) between the university, government and business sectors, will allow the formation of the cooperation network, in the search to share infrastructure and equipment that allows it to achieve common objectives in relation to the execution of professional practices.*

**H$_3$.** *The execution of the university–government–business relationship based on professional practices must have a reticular culture as well as solid managerial support to guarantee the establishment of a cooperation network.*

**H$_4$.** *The professional internship programs will define legal aspects and graduation profiles consistent with the knowledge needs of the business and government sectors.*

From the relationship between the events described, the probable scenarios are identified in Table 4.

**Table 4.** Probable scenarios.

| Scenario | H$_1$ | H$_2$ | H$_3$ | H$_4$ |
|:--------:|:-----:|:-----:|:-----:|:-----:|
| E$_1$ | 1 | 1 | 1 | 1 |
| E$_2$ | 0 | 1 | 1 | 1 |
| E$_3$ | 1 | 0 | 1 | 1 |
| E$_4$ | 1 | 1 | 0 | 1 |
| E$_5$ | 1 | 1 | 1 | 0 |
| E$_6$ | 0 | 0 | 1 | 1 |
| E$_7$ | 0 | 1 | 0 | 1 |
| E$_8$ | 0 | 1 | 1 | 0 |
| E$_9$ | 1 | 0 | 0 | 1 |
| E$_{10}$ | 1 | 0 | 1 | 0 |
| E$_{11}$ | 1 | 1 | 0 | 0 |
| E$_{12}$ | 0 | 0 | 0 | 1 |
| E$_{13}$ | 0 | 0 | 1 | 0 |
| E$_{14}$ | 0 | 1 | 0 | 0 |
| E$_{15}$ | 1 | 0 | 0 | 0 |
| E$_{16}$ | 0 | 0 | 0 | 0 |

Based on these hypotheses, the number of scenarios obeys the formula $2^n$, where $n$ is the number of hypotheses, hence 24 = 16 is the result or number of scenarios [37]. Each of these scenarios is constituted by the appearance, or not, of certain hypotheses [42]. In this case, for four hypotheses, there will be sixteen scenarios that are characterized by the occurrence or non-occurrence of these events, as presented in Table 4, where 1 is the occurrence of the event or hypothesis and 0 is the non-occurrence.

Once the hypotheses were obtained, the experts were given the SMIC survey to associate a single value per event. Such consolidation was achieved by averaging the scores obtained from each event by the experts.

- The probability of occurrence of each event was determined individually, which for the case under study is a horizon of 10 years (simple probability).
- The probability of occurrence of an event was calculated if another occurs, P(i/j), that is, the probability P, that i occurs, if J occurs (conditional probability)
- The probability of occurrence of an event was obtained, if another P(i/~j) does not occur, that is, the probability P, that i occurs, if j does not occur (conditional probability).

*5.7. Results Obtained from the Application of the SMIC Survey*

The experts consulted indicated the probability on a scale that goes from 0 to 0.9, where 0 indicates the greatest improbability and 0.9 the absolute certainty of occurrence. It is important to note that the rating scale of the probability of occurrence of the hypotheses [43] was taken, and which is reflected in Table 5.

**Table 5.** Scale of probability of occurrence of events.

| Zone | Values | Concepts |
|---|---|---|
| Zone of improbability | 0.1 | Very unlikely event |
| | 0.3 | Unlikely event |
| Doubt zone | 0.5 | Both likely and unlikely event |
| Zone of Probability | 0.7 | Likely event |
| | 0.9 | Very likely event |

Below, in Table 6, shows the set experts.

**Table 6.** Qualification of simple probability of occurrence of events P(i).

| Event | Probability | | | | |
|---|---|---|---|---|---|
| | Very Unlikely Event | Unlikely Event | Both Likely and Unlikely Event | Likely Event | Very Likely Event |
| | **0.1** | **0.3** | **0.5** | **0.7** | **0.9** |
| $E_1$ | | | X | | |
| $E_2$ | | | X | | |
| $E_3$ | | | X | | |
| $E_4$ | | | X | | |

Next, Tables 7 and 8 represent the consolidated probabilities for the data provided by the experts in terms of the probability of carrying out an event if another occurs and the probability of not carrying out an event if another occurs, respectively.

**Table 7.** Qualification of positive conditional probability of occurrence of events. P(i/j).

| Likelihood That These Events Happen | Assuming That These Are Realized | | | |
|---|---|---|---|---|
| | Event 1 | Event 2 | Event 3 | Event 4 |
| $E_1$ | | 0.808 | 0.846 | 0.732 |
| $E_2$ | 0.846 | | 0.881 | 0.831 |
| $E_3$ | 0.905 | 0.899 | | 0.820 |
| $E_4$ | 0.772 | 0.837 | 0.808 | |

**Table 8.** Qualification of negative conditional probability of occurrence of events. P(i/~j).

| Likely That These Events Happen | Assuming That They Are Not Realized | | | |
|---|---|---|---|---|
| | Event 1 | Event 2 | Event 3 | Event 4 |
| $E_1$ | | 0.207 | 0.131 | 0.309 |
| $E_2$ | 0.254 | | 0.146 | 0.232 |
| $E_3$ | 0.208 | 0.172 | | 0.278 |
| $E_4$ | 0.357 | 0.239 | 0.263 | |

The SMIC-PROB-EXPERT software (LIPSOR Laboratory) made it possible to determine the scenarios for the implementation of the proposed network, and the probability of occurrence for each of the scenarios.

Below, in Table 9, the scenarios and the probability of occurrence are presented, in the form of a matrix.

**Table 9.** Probability of scenarios.

| Scenarios | Experts |
|---|---|
| 01–1111 | 0.357 |
| 02–1110 | 0.085 |
| 03–1101 | 0.026 |
| 04–1100 | 0.003 |
| 05–1011 | 0.040 |
| 06–1010 | 0.022 |
| 07–1001 | 0.007 |
| 08–1000 | 0.016 |
| 09–0111 | 0.077 |
| 10–0110 | 0.007 |
| 11–0101 | 0.029 |
| 12–0100 | 0 |
| 13–0011 | 0.008 |
| 14–0010 | 0 |
| 15–0001 | 0.043 |
| 16–0000 | 0.278 |

By ordering the probability of occurrence, we have (see Table 10).

**Table 10.** Probability of occurrence.

| Number | Scenario | Likelihood of Occurrence | Cumulative Likelihood |
|---|---|---|---|
| 01 | 1111 | 0.357 | 0.357 |
| 16 | 0000 | 0.278 | 0.635 |
| 02 | 1110 | 0.085 | 0.720 |
| 09 | 0111 | 0.077 | 0.797 |
| 15 | 0001 | 0.043 | 0.840 |
| 05 | 1011 | 0.040 | 0.880 |
| 11 | 0101 | 0.029 | 0.909 |
| 03 | 1101 | 0.026 | 0.935 |
| 06 | 1010 | 0.022 | 0.957 |
| 08 | 1000 | 0.016 | 0.973 |
| 13 | 0011 | 0.008 | 0.981 |
| 07 | 1001 | 0.007 | 0.988 |
| 10 | 0110 | 0.007 | 0.995 |
| 04 | 1100 | 0.003 | 0.998 |
| 12 | 0100 | 0 | 0.998 |
| 14 | 0010 | 0 | 0.998 |

Visually it can be appreciated through Figure 6, the probability of occurrence of the scenarios.

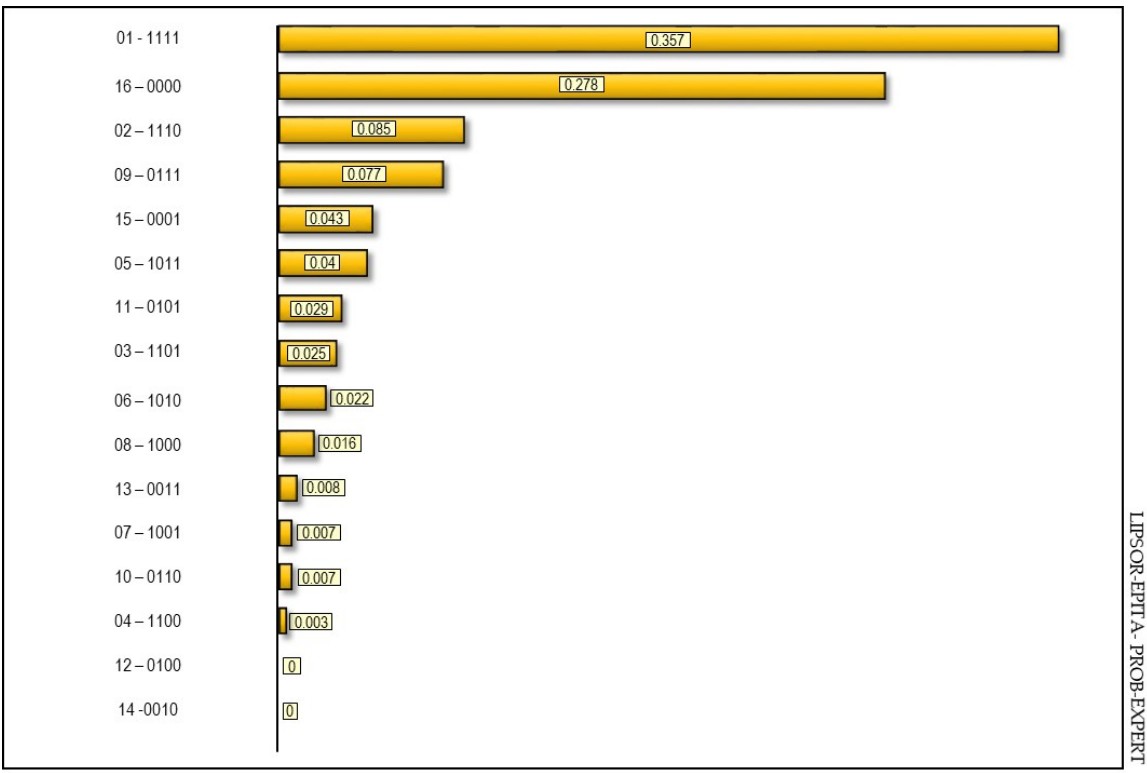

**Figure 6.** Histogram of probability of occurrence of the scenarios. Smic-Prob-Expert Software.

However, from the table and graph that record the probability of occurrence of the events, it can be deduced that the most probable scenario for the implementation of the CUGE network is scenario 01–1111, where all the events take place, their probability of occurrence is 35.7%; likewise, in general lines it can be said that the probability of occurrence of the rest of the scenarios is low, with a range that goes from 0 to 8.5% occurrence; thus, an improbability of its occurrence is inferred.

Similarly, it is important to mention that the second scenario in probability of occurrence is 16–0000, with a low probability of occurrence of 27.8%, where none of the hypotheses $H_1$, $H_2$, $H_3$, $H_4$ associated with the variables are fulfilled.

## 6. Discussion and Conclusions

Based on the results, the correspondence between the values of the motor skills–dependency relationship is determined. It is established that the variables with the greatest mobility exert a direct influence on the entire system, that is, they affect the behavior of the rest of the variables. In attention to the results, a high motricity is evidenced for variables, such as the flow of information between actors, the shared objectives and the participation of the actors in the construction of the graduation profile of university students. The importance of interpersonal communication as a strategy for achieving the objectives associated with the development of professional practices is highlighted [20,44]. In this sense, the management of the curriculum from a systemic and integrating vision represents in itself a strategy to enable sustainability in the processes of integral formation. The need for professional practices to demonstrate the development of skills for knowledge transfer is inferred, a purpose that can be promoted through cooperation networks in intersectoral and interorganizational environments, fundamentally from the establishment of strategic alliances. Therefore, cooperation represents a strategy to strengthen the sustainability of the territories [16].

The complexity in the management of university professional practices as the main category studied, as well as its relationships with the formation of intersectoral cooperation networks, have as a result a significant number of variables; however, to strengthen the levels to understand the context studied, we proceeded to reduce this system by obtaining key variables. The location of the variables by typology: input, repeaters, excluded, resulting and platoon, is an essential task to be able to analyze the motor skills–dependency relationship. The definition of key variables to describe, explain and understand the behavior of social subsystems acquires a strategic sense to the extent that it allows predicting the future behavior of a certain system [45,46] and thus specify action routes that lead to the search for the best scenario.

Within the scope of this research, the strategic axis is consolidated from the variables located in the first quadrant, where the following stand out, among others: shared objectives, reticular culture, managerial support, execution of the university–government–company relationship. The characterization of the strategic component allows defining the context studied as a dynamic, unstable and complex system, this condition is associated with the closeness in the graphic representation of some variables with respect to others. In this sense, it is required that the key variables be linked from a strategic perspective, since they reinforce each other in a systemic environment, as a basis for planning probable occurrence scenarios [47].

For the configuration of the probable and desirable scenarios, four hypotheses and their probability of occurrence was evaluated. From the hypotheses, 16 scenarios are generated. With the application of the SMIC, and the Smic-Prob-Expert software, the simple probability of occurrence of the events was obtained, and the qualification of positive and negative conditional probability of occurrence of the events. The calculation of probabilities serves as the basis for scenario planning [47], this considers trends, uncertainties, and possible ruptures. From the interaction between the actors of the system, the main changes that occur in the events with the highest probability of occurrence are identified [45]. In the present investigation, the strategic prospective leads to establishing monitoring and control in the coherence and internal consistency of the estimates that can be configured from the system of probabilities worked. The intention is to reduce the number of scenarios to specify those with the highest probability of occurrence [46].

In this sense, different occurrence scenarios are proposed for the implementation of the CUGE network, most of them have a low probability of occurrence ranging from 0 to 8.5%. The 01–1111 scenario, where all the events take place, its probability of occurrence is represented by 35.7%, being the highest in the system under study. Event 4, which refers to legal aspects and discharge profiles, is present in eight of the first thirteen possible scenarios, which indicates that it is a high-occurrence event.

The importance of prospective is perceived, from the sense of previewing to delineate trajectories of action and strategically manage organizational processes [48] as the case of professional practices in interorganizational and intersectoral cooperation contexts.

The reading to be carried out of the exposed scenarios allows us to infer that the execution of professional practices is faced with the challenge of a complex and difficult reality, which requires a great effort on the part of all the actors involved. This requires university curricula to have a high sense of flexibility and contextualization with the intention of developing professional practices where knowledge transfer is strengthened and a profile that integrates competencies, skills, values, and attitudes is consolidated [44], therefore establishing a cooperation network with effectiveness in the academic and management processes from a win–win, must combine commitment, cooperation, consensus, teamwork, and communication, which represent defining factors of the proposed network.

The role to be played by the government, in charge of mediating, sponsoring, promoting, collaborating in its creation, evolution and development of this proposal, cannot be ignored in the context of the cooperation network. The university actor present in the network is the appropriate entity to design and guide management policies within the network. The environment fostered by the built scenarios enriches the debate on critical

aspects related to the future of the organization and allows risk decision making with more transparency [49].

It is concluded that any action carried out within the studied system must pay attention to aspects such as shared objectives, modality of relationship established between the university, and other sectors (linkage, association or cooperation according to conception, nature and scope), flows communication, culture, infrastructure, legal aspects of professional practices, and graduation profile of students. Likewise, the formation of a cooperation network for the management of professional practices responds to the possibility of its articulation to the curriculum and the achievement of common objectives of the actors involved, where the previous relationships between the parties and the contributions that each one from its particularities can offer are potential factors in its consolidation.

From the arguments presented in this paper, its contribution to the study and understanding of sustainability sciences is inferred. The multidimensionality of the sustainability construct leads to the need to strengthen research that addresses a contextualized and relevant curriculum in education, where the concretion of curricular actions, such as professional practices, is the central node that allows the formation of interorganizational and intersectoral networks. The configuration of the aforementioned networks is based on cooperative relationships between different actors of the social system, this allows us to affirm that network management represents in itself a strategy that makes sustainability processes viable by strengthening local potentialities and vocations, which serve as support to consolidate development processes within the framework of the global–local paradigm. The search for strategies that contribute to the understanding of sustainability in a multidimensional perspective justifies the integration of foresight as a method of planning and management, since it allows defining trajectories that enable the formation of desirable scenarios from the conception and operationalization of networks of cooperation around professional practices and the curriculum as mobilizing variables of sustainable development.

This research can be considered a preliminary study that serves as the basis for future work where the proposed network is validated in other broader territorial areas, with similarities related to the conditions where interorganizational and intersectoral cooperation relationships occur. Because this proposal is limited to a physical geographic context of local scope, it can be considered a preliminary study for the planning, management and design of strategic action trajectories that come together in the realization of the most desirable scenarios in the system studied. The variables and events identified in this work could serve as a reference for the application in subsequent studies of other prospective techniques that, from a complementary and interdisciplinary perspective, lead to a much more complete and inclusive approach to the central categories analyzed in this investigation.

**Author Contributions:** Conceptualization, F.M.-G.; data curation, M.N.-C. and E.G.-R.; formal analysis, J.P.-G. and S.R.M.; fundraising, F.M.-G. and E.G.-R.; investigation, F.M.-G., J.P.-G. and A.S.-N.; methodology, F.M.-G.; project manager, S.R.M.; resources, A.S.-N.; software, E.G.-R.; supervision, M.N.-C.; validation, A.S.-N.; visualization, S.R.M.; writing—original draft, J.P.-G.; writing—proofreading and editing, A.S.-N. and M.N.-C. All authors have read and agreed to the published version of the manuscript.

**Funding:** This research received no external funding.

**Institutional Review Board Statement:** Not applicable for studies not involving humans or animals.

**Informed Consent Statement:** Informed consent was obtained from all subjects involved in the study.

**Data Availability Statement:** Currently, we still working in this project, and we still need to use the data for further works and analysis. However, any researcher needs the data for further investigations can contact the corresponding author via email.

**Conflicts of Interest:** The authors declare no conflict of interest.

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
