# Peer review of "Betting Scenario for the Management of University Professional Practices from the Conformation of Intersectoral Cooperation Networks"

_sustainability, doi:10.3390/su15076215_

Round 1

Reviewer 1 Report

Congratulations! I congratulate the authors for their work!

Author Response

Thank you very much for your favorable evaluation of our article.

Reviewer 2 Report

Dear authors,

The work presented here is really interesting. However, and with a view to improving the overall quality of the manuscript, the following recommendations or suggestions are made.

In this sense, first of all, it would be advisable to reinforce the theoretical framework with the greatest number of investigations on the subject published in the last three/five years. Likewise, it is suggested to use the subscripts to identify the different variables both in the development of the manuscript and in the content of the different tables. In the same way, it is advisable to check the format of all the tables and adjust it... for example, that the number of decimal places is the same in all of them and that the given nomenclature is also checked, as well as checking that the content is not broken. For example in table 3 (also appear And instead of Y).

On the other hand, it would also be recommendable to improve the structure followed for the exposition of ideas, on some occasions, it is not clear where it is going... and also perhaps at times the use of so many figures and tables in a row makes it difficult to understand the speech.

It is also suggested to optimize the conclusions with the use of current references that support or differ in the results obtained, as well as to propose and describe a future research proposal.

Author Response

Dear Reviewer:

A cordial greeting with our wishes for peace and good.

We appreciate your important recommendations because they have allowed us to significantly improve the new version of the article. Each of their recommendations were addressed, as stated below and can be evidenced in the corresponding file:

The work presented here is really interesting. However, and with a view to improving the overall quality of the manuscript, the following recommendations or suggestions are made.

In this sense, first of all, it would be advisable to reinforce the theoretical framework with the greatest number of investigations on the subject published in the last three/five years.

A background of the study section was prepared that includes important citations from recently published high-impact scientific literature, this is evidenced in the section:

  1. Background of the Study

Numerous investigations have been carried out with the purpose of studying the activities of companies in cooperation networks in coherence with the ideas of [15-16 ] the competences of social innovation agents, to operate jointly require that the organizations and among them universities generate intersectoral spaces that are configured in cooperation networks around common objectives capable of operationally connecting the different sectors of activity in society and specifically in the context of management of professional practices of the students [17], which occur mainly in the productive sector [18]. This relationship that arises in the academic context generates strong and lasting links as well as alliances among the various sectors that are intertwined in the training processes of professional practice in universities, enabling the formation of intersectoral cooperation networks [ 16].

In coherence with the perspective of [19] who developed a research on the organizational DNA for the quality of service in public universities, they affirm that the development of human talent must be oriented to the performance of alternatives and innovations in the different universities from the production and application of knowledge. From the perspective of [9], the creation of intersectoral knowledge networks favors the development of the productive, government and university sectors, reduces uncertainty and promotes the social capital of the locality, which contributes to decision-making in the various sectors of activity, while strengthening relationships between them, [20].

This finding is consistent with the results of [21], interaction between cultural, social, economic and political factors leads to the emergence of complex problems [22]. Vast amount of evidence calls for a new solution that cannot be found by a single actor from one sector; In this sense, it states that the interactions and innovations in the University, government and productive sector spheres behave like a triple helix [23-24] that opposes creative knowledge gaps to solve problems in the government or productive sector, a path different from the linear technocratic one, but rather a disruptive path that involves alliances and cooperative actions in the network as a space where innovations are generated to solve real problems in different sectors of society [25-26]; as well as the promotion of the establishment and/or consolidation of strategic ties between governments, the productive sector, civil society organizations and institutions of university education, science and technology.[27-28].

From these perspectives, management in higher education is assumed as an intentional process organized and oriented towards the optimization of operations, procedures and internal projects of higher education institutions, with the aim of approaching the purposes for which they were created. the institutions; which implies perfecting the pedagogical, managerial, community and administrative procedures that are mobilized in it, to achieve this it is necessary to have the competition and joint work of all interested parties. [29-30].

The actors in the educational field, to obtain successful and innovative results, require strengthening the capacities to cooperate and establish lasting alliances and agreements that make it possible to project, design, analyze and evaluate the processes of labor and disciplinary training of students in space of professional practices, so that the construction and transfer of knowledge that occurs in academic and work spaces can become innovations that generate social welfare to the extent that they respond to the demands of the environment [31].

In this framework of ideas, the management processes of education systems require attention to factors such as planning, equity, quality, resource management, community participation through a good governance model that guarantee the governance and governability of educational institutions, generate trust and credibility from accountability to the Society; all this, to generate optimal results and the provision of better services; among them, the management of professional practices.

Likewise, it is suggested to use the subscripts to identify the different variables both in the development of the manuscript and in the content of the different tables.

The subindices were used in the names of the variables; It is evidenced in the corresponding tables, for example:

X1: Legal aspects related to professional practices.

In the same way, it is advisable to check the format of all the tables and adjust it... for example, that the number of decimal places is the same in all of them and that the given nomenclature is also checked, as well as checking that the content is not broken. For example in table 3 (also appear And instead of Y).

The format of all the tables was checked and the indicated adjustments were made (see tables in the new version of the manuscript), likewise, the number of decimals was adjusted so that it is the same in all the figures and the nomenclature was adjusted according to the indications given (see tables 1, 2 and 3)

On the other hand, it would also be recommendable to improve the structure followed for the exposition of ideas, on some occasions, it is not clear where it is going... and also perhaps at times the use of so many figures and tables in a row makes it difficult to understand the speech.

The coherence of the ideas was reviewed and by reorganizing the sequentiality and logic, for example in the introduction, a better understanding of the message to be transmitted is achieved. The sequentiality in the tables and figures was also reviewed, introducing a text between them that allows a better understanding, for example:

Below, in table 6, shows the simple probabilities assigned to each event by the set of experts.

It is also suggested to optimize the conclusions with the use of current references that support or differ in the results obtained, as well as to propose and describe a future research proposal.

A new section was elaborated that integrates the discussion and the conclusions; This section is in line with the objectives of the research and allows theoretical verification with recent mainstream scientific literature, as evidenced below:

  1. Discussion and conclusions:

Based on the results, the correspondence between the values of the motor skills - dependency relationship is determined. It is established that the variables with the greatest mobility exert a direct influence on the entire system, that is, they affect the behavior of the rest of the variables. In attention to the results, a high motricity is evidenced for variables such as: The flow of information between actors, the shared objectives and the participation of the actors in the construction of the graduation profile of university students. The importance of interpersonal communication as a strategy for achieving the objectives associated with the development of professional practices is highlighted [44-45]. In this sense, the management of the curriculum from a systemic and integrating vision represents in itself a strategy to enable sustainability in the processes of integral formation. The need for professional practices to demonstrate the development of skills for knowledge transfer is inferred, a purpose that can be promoted through cooperation networks in intersectoral and interorganizational environments, fundamentally from the establishment of strategic alliances. Therefore, cooperation represents a strategy to strengthen the sustainability of the territories [46].

The complexity in the management of university professional practices as the main category studied, as well as its relationships with the formation of intersectoral cooperation networks have as a result a significant number of variables, however, to strengthen the levels to understand the context studied, we proceeded to reduce this system by obtaining key variables. The location of the variables by typology: input, repeaters, excluded, resulting and platoon is an essential task to be able to analyze the motor skills - dependency relationship. The definition of key variables to describe, explain and understand the behavior of social subsystems acquires a strategic sense to the extent that it allows predicting the future behavior of a certain system. [47-48] and thus specify action routes that lead to the search for the bet scenario.

Within the scope of this research, the strategic axis is consolidated from the variables located in the first quadrant, where the following stand out, among others: shared objectives, reticular culture, managerial support, execution of the university-government-company relationship. The characterization of the strategic component allows defining the context studied as a dynamic, unstable and complex system, this condition is associated with the closeness in the graphic representation of some variables with respect to others. In this sense, it is required that the key variables be linked from a strategic perspective, since they reinforce each other in a systemic environment, as a basis for planning probable occurrence scenarios [49].

For the configuration of the probable and desirable scenarios, four hypothesis and its probability of occurrence was evaluated. From the hypotheses, 16 scenarios are generated. With the application of the SMIC, and the Smic-Prob-Expert software, the simple probability of occurrence of the events was obtained, and the qualification of positive and negative conditional probability of occurrence of the events. The calculation of probabilities serves as the basis for scenario planning (Labrín-Mesía and Ruiz-Ruiz, 2022), this considers trends, uncertainties and possible ruptures. From the interaction between the actors of the system, the main changes that occur in the events with the highest probability of occurrence are identified [47]. In the present investigation, the strategic prospective leads to establishing monitoring and control in the coherence and internal consistency of the estimates that can be configured from the system of probabilities worked. The intention is to reduce the number of scenarios to specify those with the highest probability of occurrence [48].

In this sense, different occurrence scenarios are proposed for the implementation of the CUGE network, most of them have a low probability of occurrence ranging from 0 to 8.5%. The 01-1111 scenario, where all the events take place, its probability of occurrence is represented by 35.7%, being the highest in the system under study. Event 4, which refers to legal aspects and discharge profiles, is present in eight of the first thirteen possible scenarios, which indicates that it is a high-occurrence event.

The importance of prospective is perceived, from the sense of previewing to delineate trajectories of action and strategically manage organizational processes [50] as the case of professional practices in interorganizational and intersectoral cooperation contexts.

The reading to be carried out of the exposed scenarios allows us to infer that the execution of professional practices is faced with the challenge of a complex and difficult reality, which requires a great effort on the part of all the actors involved; This requires university curricula to have a high sense of flexibility and contextualization with the intention of developing professional practices where knowledge transfer is strengthened and a profile that integrates competencies, skills, values, and attitudes is consolidated [44], therefore, establish a Cooperation Network with effectiveness in the academic and management processes from a win-win, must combine: commitment, cooperation, consensus, teamwork, communication; which represent defining factors of the proposed network.

The role to be played by the government, in charge of mediating, sponsoring, promoting, collaborating in its creation, evolution and development of this proposal, cannot be ignored in the context of the cooperation network. The university actor present in the network is the appropriate entity to design and guide management policies within the network. The environment fostered by the built scenarios enriches the debate on critical aspects related to the future of the organization and allows risk decision-making with more transparency [51].

It is concluded that any action carried out within the studied system must pay attention to aspects such as: shared objectives, modality of relationship established between the university and other sectors (linkage, association or cooperation according to conception, nature and scope), flows communication, culture, infrastructure, legal aspects of professional practices, graduation profile of students. Likewise, the formation of a cooperation network for the management of professional practices responds to the possibility of its articulation to the curriculum and the achievement of common objectives of the actors involved, where the previous relationships between the parties and the contributions that each one from its particularities can offer are potential factors in its consolidation.

In the final paragraph of the discussion and conclusions section, some lines of future research are identified.

This research can be considered a preliminary study that serves as the basis for future work where the proposed network is validated in other broader territorial areas, with similarities related to the conditions where interorganizational and intersectoral cooperation relationships occur. Because this proposal is limited to a physical geographic context of local scope, it can be considered a preliminary study for the planning, management and design of strategic action trajectories that come together in the realization of the most desirable scenarios in the system studied. The variables and events identified in this work could serve as a reference for the application in subsequent studies of other prospective techniques that, from a complementary and interdisciplinary perspective, lead to a much more complete and inclusive approach to the central categories analyzed in this investigation.

New citations and references can be identified in the new version of the article underlined in red.

Thank you very much for your contributions.

The authors

Reviewer 3 Report

It is an interesting topic; however, there are some significant problems with the paper.

·       Instead of beginning with the purpose of the study in the first paragraph, the authors may introduce the constructs then they can give the purpose, research gap, and their contribution.

·         The paragraph between lines 64-87 is too long. There are some different opinions in the same paragraph. This paragraph can be divided into 2-3 paragraphs.

·         In the introduction, the gap, originality, and contribution should be discussed fluently.

·         Do we really need Fig.1? I think it is not needed.

·         I think a theoretical basement is missing because the paper directly goes to methodology after the introduction.

·         Where did you get/find Table 1 variables? How did you define them?

·         The results depend on seven experts’ subjective evaluations. However, the problem is the validity and reliability of the evaluations of the experts. I don’t see any attempt at the validity and reliability of the evaluations.

·         The details in Fig.3 cannot be read. I don’t understand the figure.

·         What are the four zones in Fig.4? What do they represent? They don’t explain anything to a reader who does not know the MICMAC program.

·         There is no discussion of the findings. The results should be connected to the literature.

·         What are the contributions of the study to the existing theory and higher education managers?

Author Response

Dear Reviewer:

A cordial greeting with our wishes for peace and good.

We appreciate your important recommendations because they have allowed us to significantly improve the new version of the article. Each of their recommendations were addressed, as stated below and can be evidenced in the corresponding file:

It is an interesting topic; however, there are some significant problems with the paper.

  •      Instead of beginning with the purpose of the study in the first paragraph, the authors may introduce the constructs then they can give the purpose, research gap, and their contribution

In response to this indication, the introduction was organized from a new logic and sequentiality as evidenced below:

Strategic prospective is considered by [1], as a formalized approach that uses a combination of qualitative and quantitative tools, to obtain multiple scenarios of future alternatives, supported by comprehensive analysis and incorporating possible actions and their consequences. In this regard, [2] refer that it consists of analyzing the phenomena from a global, systemic and dynamic perspective, in order to generate future scenarios, built not only based on past data but also taking into account the future evolution of the variables, as well as the behavior of the actors involved, so as to minimize uncertainty and lead the present action towards an acceptable or desired future. In the particular case of strategic prospective, scenarios are proposed as a tool that contributes to the analysis of opportunities and threats in the environment and thus describe possible futures in correspondence with a certain problematic situation [3] (p. 9). Noting that for [4] it is not enough that the foreseen futures can occur, it is also necessary that the social actors involved are able to make at least one of them a reality.

In this regard, [5] refer that strategic prospective should not be understood as a prediction or forecast of tomorrow's events, but as an analysis of the future, to try to understand it and influence its dynamics. The construction of scenarios reduces uncertainty regarding the future behavior of a certain social phenomenon, which allows carrying out better planning and investment processes [6]; in addition to facilitating the actors to participate in the construction of the desired future [7].

From the ideas exposed, it is inferred that the future can be estimated and thus provide a greater degree of certainty with respect to its trajectories of action; therefore, organizations seek to define and implement strategies to achieve the desired futures.

The main purpose of the article is to build an ideal scenario where an intersectoral cooperation network can be operationalized that makes possible the effective management of professional practices developed by university students in their last years of academic training, intervening as an empirical context for research in the Paraguaná Peninsula, Falcón state, Venezuela.

Regarding the prospective to estimate the impact associated with the operation of an intersectoral cooperation network, (University-company-students) it could contribute to universities developing improvements in their management mechanisms, educational quality processes, obtaining competitive advantages in their mission functions, strengthen relationships with their environment, as well as assess the relevance of the profile of graduates, among other descriptors. In relation to companies, improvement in their characteristic processes and products, training of their employees is expected; likewise, innovation and solutions to context problems would be encouraged. Concerning students, they would show improvements in their training processes from a complex and transdisciplinary vision, based on theory-practice integration; the purpose is to build significant learning experiences that correspond to the dynamics of the world of work, as a basis for developing professional and occupational skills.

In this regard [8], state that the importance of cooperation lies in the fact that it contributes to reinforcing the competitive position of companies and, consequently, of their territories. In this same order [9], they refer to the improvement of the standard of living of localities and rural or urban territories from the cooperation between the social sectors.

From the cooperation networks, a greater increase in the accessibility of resources, a greater amount of knowledge and shared information, the creation of new product lines and a greater competitive consolidation are promoted [10]. In this regard, [11] emphasizes the transfer of knowledge as a challenge for educational organizations.

The consolidation of networks is inferred as a way to strengthen transaction and exchange processes. Taken into consideration [12] when they consider that the driving factors of network cooperation multiply as a network evolves in time and space, and the factors that influence it become increasingly diverse. In this sense, authors such as [13-14], highlight the importance of educational prospective and network design, as key strategic processes for growth and development in organizational, interorganizational, and intersectoral fields.

Consequently, in this article, the search for the future places the reader from the dynamics of the intersectoral context that is the object of the investigation, then the methodological systematization is presented to later organize the results, their analysis and interpretation in light of the theoretical references.

      The paragraph between lines 64-87 is too long. There are some different opinions in the same paragraph. This paragraph can be divided into 2-3 paragraphs.  In the introduction, the gap, originality, and contribution should be discussed fluently.

The division indicated for the paragraph was made and thus a better understanding of the message to be transmitted to the reader is achieved.

By reorganizing the logic and sequentiality and delving into the problem situation, an answer is given to this indication, for example:

From the cooperation networks, a greater increase in the accessibility of resources, a greater amount of knowledge and shared information, the creation of new product lines and a greater competitive consolidation are promoted [10]. In this regard, [11] emphasizes the transfer of knowledge as a challenge for educational organizations.

The consolidation of networks is inferred as a way to strengthen transaction and exchange processes. Taken into consideration [12] when they consider that the driving factors of network cooperation multiply as a network evolves in time and space, and the factors that influence it become increasingly diverse. In this sense, authors such as [13-14], highlight the importance of educational prospective and network design, as key strategic processes for growth and development in organizational, interorganizational, and intersectoral fields.

Consequently, in this article, the search for the future places the reader from the dynamics of the intersectoral context that is the object of the investigation, then the methodological systematization is presented to later organize the results, their analysis and interpretation in light of the theoretical references.

       Do we really need Fig.1? I think it is not needed.

This figure was removed.

  • I think a theoretical basement is missing because the paper directly goes to methodology after the introduction.

A section is included to address this indication, as shown below:

  1. Background of the Study

Numerous investigations have been carried out with the purpose of studying the activities of companies in cooperation networks in coherence with the ideas of [15-16 ] the competences of social innovation agents, to operate jointly require that the organizations and among them universities generate intersectoral spaces that are configured in cooperation networks around common objectives capable of operationally connecting the different sectors of activity in society and specifically in the context of management of professional practices of the students [17], which occur mainly in the productive sector [18]. This relationship that arises in the academic context generates strong and lasting links as well as alliances among the various sectors that are intertwined in the training processes of professional practice in universities, enabling the formation of intersectoral cooperation networks [ 16].

In coherence with the perspective of [19] who developed a research on the organizational DNA for the quality of service in public universities, they affirm that the development of human talent must be oriented to the performance of alternatives and innovations in the different universities from the production and application of knowledge. From the perspective of [9], the creation of intersectoral knowledge networks favors the development of the productive, government and university sectors, reduces uncertainty and promotes the social capital of the locality, which contributes to decision-making in the various sectors of activity, while strengthening relationships between them, [20].

This finding is consistent with the results of [21], interaction between cultural, social, economic and political factors leads to the emergence of complex problems [22]. Vast amount of evidence calls for a new solution that cannot be found by a single actor from one sector; In this sense, it states that the interactions and innovations in the University, government and productive sector spheres behave like a triple helix [23-24] that opposes creative knowledge gaps to solve problems in the government or productive sector, a path different from the linear technocratic one, but rather a disruptive path that involves alliances and cooperative actions in the network as a space where innovations are generated to solve real problems in different sectors of society [25-26]; as well as the promotion of the establishment and/or consolidation of strategic ties between governments, the productive sector, civil society organizations and institutions of university education, science and technology.[27-28].

From these perspectives, management in higher education is assumed as an intentional process organized and oriented towards the optimization of operations, procedures and internal projects of higher education institutions, with the aim of approaching the purposes for which they were created. the institutions; which implies perfecting the pedagogical, managerial, community and administrative procedures that are mobilized in it, to achieve this it is necessary to have the competition and joint work of all interested parties. [29-30].

The actors in the educational field, to obtain successful and innovative results, require strengthening the capacities to cooperate and establish lasting alliances and agreements that make it possible to project, design, analyze and evaluate the processes of labor and disciplinary training of students in space of professional practices, so that the construction and transfer of knowledge that occurs in academic and work spaces can become innovations that generate social welfare to the extent that they respond to the demands of the environment [31].

In this framework of ideas, the management processes of education systems require attention to factors such as planning, equity, quality, resource management, community participation through a good governance model that guarantee the governance and governability of educational institutions, generate trust and credibility from accountability to the Society; all this, to generate optimal results and the provision of better services; among them, the management of professional practices.

Where did you get/find Table 1 variables? How did you define them? The results depend on seven experts’ subjective evaluations. However, the problem is the validity and reliability of the evaluations of the experts. I don’t see any attempt at the validity and reliability of the evaluations.

This concern is addressed through the following text:

It is highlighted that the study variables emerged from the transit of a descriptive diagnosis in relation to the input knowledge of existing theories, such as social network theories, cooperation theories, university relationship models with the external sector. (company, government), as well as the university curriculum regarding the management of professional practices; The treatment that some authors make to certain categories considered main in the investigation was revealed, in this sense, scientific articles, books, university curriculum, doctoral theses in relation to the investigated topic were analyzed, which allowed the generation of a questionnaire of closed questions with Likert-type scale which was applied to the key actors of the study, prior to a validation process through expert judgment and reliability determined through Cronbach's Alpha coefficient.

      The details in Fig.3 cannot be read. I don’t understand the figure.

The figure was improved.

       What are the four zones in Fig.4? What do they represent? They don’t explain anything to a reader who does not know the MICMAC program.

This indication is addressed through the following text:

... In this regard, the power zone is characterized by high motricity and the lowest dependency, consequently, the variables found in this zone are the most important in the system since they influence the vast majority and depend little on them. Any modification that occurs in it affects the entire system; the autonomous zone hosts the excluded variables, those that neither influence others nor are they influenced by the other variables of the system; the Conflict Zone, receives its name due to the great instability shown by the variables that converge there, due to its high mobility and high dependency; The Exit Zone, characterized by having low mobility, but high dependency, groups the so-called result variables, as a consequence of the impact of the influence of the variables of the zone of power and conflict.

       There is no discussion of the findings. The results should be connected to the literature.

A discussion and conclusions section is included that allows comparison with the recent mainstream scientific literature, as evidenced below:

  1. Discussion and conclusions:

Based on the results, the correspondence between the values of the motor skills - dependency relationship is determined. It is established that the variables with the greatest mobility exert a direct influence on the entire system, that is, they affect the behavior of the rest of the variables. In attention to the results, a high motricity is evidenced for variables such as: The flow of information between actors, the shared objectives and the participation of the actors in the construction of the graduation profile of university students. The importance of interpersonal communication as a strategy for achieving the objectives associated with the development of professional practices is highlighted [44-45]. In this sense, the management of the curriculum from a systemic and integrating vision represents in itself a strategy to enable sustainability in the processes of integral formation. The need for professional practices to demonstrate the development of skills for knowledge transfer is inferred, a purpose that can be promoted through cooperation networks in intersectoral and interorganizational environments, fundamentally from the establishment of strategic alliances. Therefore, cooperation represents a strategy to strengthen the sustainability of the territories [46].

The complexity in the management of university professional practices as the main category studied, as well as its relationships with the formation of intersectoral cooperation networks have as a result a significant number of variables, however, to strengthen the levels to understand the context studied, we proceeded to reduce this system by obtaining key variables. The location of the variables by typology: input, repeaters, excluded, resulting and platoon is an essential task to be able to analyze the motor skills - dependency relationship. The definition of key variables to describe, explain and understand the behavior of social subsystems acquires a strategic sense to the extent that it allows predicting the future behavior of a certain system. [47-48] and thus specify action routes that lead to the search for the bet scenario.

Within the scope of this research, the strategic axis is consolidated from the variables located in the first quadrant, where the following stand out, among others: shared objectives, reticular culture, managerial support, execution of the university-government-company relationship. The characterization of the strategic component allows defining the context studied as a dynamic, unstable and complex system, this condition is associated with the closeness in the graphic representation of some variables with respect to others. In this sense, it is required that the key variables be linked from a strategic perspective, since they reinforce each other in a systemic environment, as a basis for planning probable occurrence scenarios [49].

For the configuration of the probable and desirable scenarios, four hypothesis and its probability of occurrence was evaluated. From the hypotheses, 16 scenarios are generated. With the application of the SMIC, and the Smic-Prob-Expert software, the simple probability of occurrence of the events was obtained, and the qualification of positive and negative conditional probability of occurrence of the events. The calculation of probabilities serves as the basis for scenario planning (Labrín-Mesía and Ruiz-Ruiz, 2022), this considers trends, uncertainties and possible ruptures. From the interaction between the actors of the system, the main changes that occur in the events with the highest probability of occurrence are identified [47]. In the present investigation, the strategic prospective leads to establishing monitoring and control in the coherence and internal consistency of the estimates that can be configured from the system of probabilities worked. The intention is to reduce the number of scenarios to specify those with the highest probability of occurrence [48].

In this sense, different occurrence scenarios are proposed for the implementation of the CUGE network, most of them have a low probability of occurrence ranging from 0 to 8.5%. The 01-1111 scenario, where all the events take place, its probability of occurrence is represented by 35.7%, being the highest in the system under study. Event 4, which refers to legal aspects and discharge profiles, is present in eight of the first thirteen possible scenarios, which indicates that it is a high-occurrence event.

The importance of prospective is perceived, from the sense of previewing to delineate trajectories of action and strategically manage organizational processes [50] as the case of professional practices in interorganizational and intersectoral cooperation contexts.

The reading to be carried out of the exposed scenarios allows us to infer that the execution of professional practices is faced with the challenge of a complex and difficult reality, which requires a great effort on the part of all the actors involved; This requires university curricula to have a high sense of flexibility and contextualization with the intention of developing professional practices where knowledge transfer is strengthened and a profile that integrates competencies, skills, values, and attitudes is consolidated [44], therefore, establish a Cooperation Network with effectiveness in the academic and management processes from a win-win, must combine: commitment, cooperation, consensus, teamwork, communication; which represent defining factors of the proposed network.

The role to be played by the government, in charge of mediating, sponsoring, promoting, collaborating in its creation, evolution and development of this proposal, cannot be ignored in the context of the cooperation network. The university actor present in the network is the appropriate entity to design and guide management policies within the network. The environment fostered by the built scenarios enriches the debate on critical aspects related to the future of the organization and allows risk decision-making with more transparency [51].

It is concluded that any action carried out within the studied system must pay attention to aspects such as: shared objectives, modality of relationship established between the university and other sectors (linkage, association or cooperation according to conception, nature and scope), flows communication, culture, infrastructure, legal aspects of professional practices, graduation profile of students. Likewise, the formation of a cooperation network for the management of professional practices responds to the possibility of its articulation to the curriculum and the achievement of common objectives of the actors involved, where the previous relationships between the parties and the contributions that each one from its particularities can offer are potential factors in its consolidation.

This research can be considered a preliminary study that serves as the basis for future work where the proposed network is validated in other broader territorial areas, with similarities related to the conditions where interorganizational and intersectoral cooperation relationships occur. Because this proposal is limited to a physical geographic context of local scope, it can be considered a preliminary study for the planning, management and design of strategic action trajectories that come together in the realization of the most desirable scenarios in the system studied. The variables and events identified in this work could serve as a reference for the application in subsequent studies of other prospective techniques that, from a complementary and interdisciplinary perspective, lead to a much more complete and inclusive approach to the central categories analyzed in this investigation.

      what are the contributions of the study to the existing theory and higher education managers?

This indication is answered in the last paragraph of the Background of the Study:

         In this framework of ideas, the management processes of education systems require attention to factors such as planning, equity, quality, resource management, community participation through a good governance model that guarantee the governance and governability of educational institutions, generate trust and credibility from accountability to the Society; all this, to generate optimal results and the provision of better services; among them, the management of professional practices.

New citations and references can be identified in the new version of the article underlined in red.

Thank you very much for your contributions.

The authors

Round 2

Reviewer 2 Report

Dear authors, 

Congratulations for your efforts. The manuscript have improved considerably. The only thing that should be reviewed is the table number 1. The title is in Spanish and the format is moved.

Thanks in advances.

Author Response

Thank you very much, your contributions have been essential to obtain a better quality product.

Reviewer 3 Report

The paper has been improved and necessary changes have been applied.

Author Response

(The authors gave the same response as above.)
